# Investigation Methods for Vocal Onset—A Historical Perspective

**DOI:** 10.3390/bioengineering11100989

**Published:** 2024-09-30

**Authors:** Bernhard Richter, Matthias Echternach, Louisa Traser

**Affiliations:** 1Institute for Musicians’ Medicine, Faculty of Medicine, Freiburg University, and Freiburg University Medical Center, Elsässer Str. 2m, 79110 Freiburg, Germany; louisa.traser@uniklinik-freiburg.de; 2Division Phoniatrics and Pediatric Audiology, Department of Otolaryngology, Munich University Hospital, and Faculty of Medicine, Munich University (LMU), Marchioninistraße 15, 81377 München, Germany; matthias.echternach@med.uni-muenchen.de

**Keywords:** voice physiology, vocal onset, high-speed video laryngoscopy, ultra-fast three-dimensional MRI

## Abstract

The topic of phonation onset gestures is of great interest to singers, acousticians, and voice physiologists alike. The vocal pedagogue and voice researcher Manuel Garcia, in the mid-19th century, first coined the term “coup de la glotte”. Given that Garcia defined the process as “a precise articulation of the glottis that leads to a precise and clean tone attack”, the term can certainly be linked to the concept of “vocal onset” as we understand it today. However, Garcia did not, by any means, have the technical measures at his disposal to investigate this phenomenon. In order to better understand modern ways of investigating vocal onset—and the limitations that still exist—it seems worthwhile to approach the subject from a historical perspective. High-speed video laryngoscopy (HSV) can be regarded as the gold standard among today’s examination methods. Nonetheless, it still does not allow the three-dimensionality of vocal fold vibrations to be examined as it relates to vocal onset. Clearly, measuring methods in voice physiology have developed fundamentally since Garcia’s time. This offers grounds for hope that the still unanswered questions around the phenomenon of vocal onset will be resolved in the near future. One promising approach could be to develop ultra-fast three-dimensional MRI further.

## 1. Introduction

Vocal onset is the process occurring between the first detectable oscillatory glottal movement and the steady vibration of the vocal folds [1]. In singing and speaking, this phenomenon occurs during the transition from expiration to phonation, after short pauses and during the transition from unvoiced to voiced sounds [2]. Onset processes are of acoustic significance for timbre. The duration and course of this transient process depend on the excitation of the resonance system and the damping. The onset process can be significantly influenced by vocal fold tension and subglottic pressure. There are a variety of parameters by which to quantify the processes when phonation commences. The individual parameters are differentiated by highly varied measurement methods, including visual depiction in high-speed video laryngoscopy (HSV).

From a modern perspective, a scientific, evidence-based approach to the topic of vocal onset that makes use of sophisticated technical possibilities seems self-evident. However, if we look back to the time of the vocal pedagogue and voice researcher Manuel Garcia (1805–1906), who introduced the term “coup de la glotte” into vocal training terminology in the mid-19th century, no firmly established tradition existed, either in terms of scientific discourse or technical possibilities for investigating this phenomenon. Since Garcia defines the process as “a precise articulation of the glottis, which results in a precise and clean tone onset”, a connection to the term “vocal onset” can certainly be made conceptually [3]. Ever since Garcia’s first description, researchers from various disciplines have given much thought to how to analyze the exact beginning of a sound [4,5,6,7].

The following begins with a brief excursus on the evolution of a scientific way of thinking, leading into a discussion of the development of the general technical and laryngological prerequisites for modern-day vocal physiology research. Finally, the focus is turned to an investigation of vocal onset.

### 1.1. From Belief to Knowledge

The Latin-speaking academic world of the Middle Ages developed a research and teaching methodology based on the logical writings and understanding of science of the Greek philosopher Aristotle (384–322 BC). This aimed to clarify questions in university teaching by means of theoretical considerations based on dogmatic premises, and later became known as “scholasticism”. For example, in 1499, Johann Peyligk (1474–1512) repeated Aristotle’s opinion in a standard medical work—without any anatomical or physiological investigations of his own—that the voice is formed in the same way as in a pipe [8]. Resistance to this tradition, with its theoretical book knowledge and lack of practical relevance, grew during the Renaissance. Around 1700, with the Age of Enlightenment, European thought began to increasingly distance itself from the doctrines and beliefs of the Middle Ages.

The first anatomical drawings depicting the construction of the larynx with its most important structures including the vocal folds were made as early as the Renaissance by Leonardo da Vinci (1452–1519), though without an understanding of how the individual elements worked [9]. Subsequently, the Italian anatomists Vesalius (1514–1564), Fallopio (1523–1563), and Eustachius (1520–1574), in particular, considerably refined our knowledge of the cartilage, muscles, and nerves of the larynx, without, however, arriving at any reliable findings concerning voice production. Other anatomists of the 16th and 17th centuries such as Fabricius (1537–1619) and Casserius (1561–1616) considered the development of the voice in the larynx, but likewise did not identify the precise mechanism whereby sound is produced.

At the beginning of the 18th century, Denis Dodart (1634–1707), a French physician and professor at the Faculty of Medicine at the University of Paris, deduced that the production of sound in the larynx could be explained in the same way as whistling sounds generated at the lips of the mouth [10].

In medicine, physiology increasingly became the focus of interest. Physiologists make observations about physical processes and formulated hypotheses based on the conclusions resulting from these observations, which they tested by means of experiments. Insights gained in this way included the first description in 1741 of the voice production mechanisms in the larynx by Antoine Ferrein (1693–1769) [11], a significant advance in terms of vocal physiology. These early physiological experiments can be seen as preparation for the paradigm shift to the scientific evidence-based medicine of modern times [12]. This new knowledge-based approach inspired several scientists to approach the question of how and where the voice was produced, and how vowels were formed in speech and singing, by constructing machines that attempted to imitate human voice production and, in particular, the formation of vowels. Three scientists made significant discoveries in this regard: Christian Gottlieb Kratzenstein in 1780 [13], Wolfgang von Kempelen in 1791 [14], and Johan Nepomuk Maelzel in 1824 [15]. In the arts, the writer E. T. A. Hoffmann (1776–1822) dealt with this theme of “man as machine” in his 1816 novella *Der Sandmann.* This work features the character of Olympia, who was in turn brought to the stage by Jacques Offenbach (1819–1880) in his opera *Les Contes d’Hoffmann* at the Paris Opéra comique in 1881.

This deep desire to reproduce the human voice in models continued into the 20th and 21st centuries. Johan Sundberg and his colleagues developed a model based on X-ray images of the vocal tract, named Apex, in which the acoustic influences of the tongue position, jaw opening, and lip position can be reproduced [16]. In addition, his team also developed a voice synthesizer called Madde [17], in which overtones, formants, and vibrato can be varied. New technology enabled the signal of the voice source to be computer-generated in these models.

### 1.2. General Technical Prerequisites

Voice production is based on mechanical principles that are very fast. In humans, the frequency of vocal fold vibrations ranges from just above 50 Hz to well over 1500 Hz. At medium-speaking pitch, the vibrations of the vocal folds lie between approx. 100 and 250 Hz. In singing, Mozart’s operas can be used as examples to vividly illustrate this range in classical singing. If the concert pitch A4 is standardized at 440 Hz, then the bass who sings the role of Osmin in “The Abduction from the Seraglio” sings at the lowest note D2 (=73 Hz), while the Queen of the Night in “The Magic Flute” sings as her highest note F6 (1397 Hz). In non-classical singing, significantly higher frequencies are regularly reached. The Guinness Book of Records lists values of more than 5000 Hz for the highest frequencies that can be produced by the human voice [18].

The frequencies of vocal fold vibrations during singing and speaking—as well as vocal onset—are therefore far too high for the time resolution of the human eye, well above the flicker fusion threshold of human vision of around 50 Hz. In order to visualize vocal fold vibrations in general and vocal onset in particular, special technology for slowing down the vibration sequences is required.

The first method for slowing down the depiction of movement, the so-called stroboscopic effect, was described by the English physician Peter Mark Roget (1779–1869), who published his observations and calculations on this phenomenon in 1825 [19]. The stroboscopic principle was first termed as such by one of the earliest users, Simon Ritter von Stampfer (1790–1864), in 1832. Von Stampfer created a motion display using a rotating disk with slots, which was called “Prof. Stampfer’s stroboscopic disk” or the “optical magic disk” [20].

To allow these effects to be useful in scientific research, the technical requirements of electrical engineering had to be developed as the essential basis. Photographic methods and film were also necessary. These developments began as early as the 19th century. In 1800, Alessandro Volta (1745–1827) invented the first battery [21], while Joseph Nicéphore Niépce (1765–1833) succeeded in taking the first photograph in 1826 [22]. The year 1895 saw the first public film screenings by the brothers Auguste and Louis Lumière in Paris [23]. The rapid development of sound transmission and recording through the invention of the telephone and the gramophone and the development of microphones and broader field recording technology were also very significant around this time. The outstanding importance of these inventions for the “art of sound” is also underlined by the fact that Alexander Graham Bell (1847–1922), one of the most important personalities in this field, was given the honor of having the unit of measurement decibel named after him [24].

The combination of sound and image, first achieved about thirty years later in a so-called “talkie”, subsequently enabled the development of highly valuable visualization methods for voice research and, in particular, the documentation and reproducibility of movement sequences during voice production. Interestingly, the first commercially successful “sound motion picture”—*The Jazz Singer* in 1927—concerned a singer’s career.

Another important invention in the history of science is the method that made it possible to look inside the (living) human body. For this, credit must go to Conrad Wilhelm Röntgen (1845–1923), who discovered X-rays in 1895 [25]. A second procedure, of even greater significance for voice research today, is magnetic resonance imaging, as first described in 1973 by Paul Lauterbur [26].

### 1.3. Laryngological Requirements

#### 1.3.1. Basics

After Garcia first described the use of a laryngeal mirror to observe the vocal folds in the larynx during phonation in 1854 [27], Max Joseph Oertel (1835–1897) was the first to use the stroboscopic technique in laryngology in 1878 [28]. Albert Musehold (1854–1919) was innovative in his combination of stroboscopy and photography in individual voice physiology studies [29]. From the middle of the 20th century onwards, stroboscopy was also employed for the diagnosis of voice disorders, as described by Beck and Schönhärl in 1958 [30]. Volker Barth (1943–2011) succeeded in integrating stroboscopy into magnifying laryngoscopy, thus achieving a visualization of the vocal folds that was not only slowed-down but optically enlarged, too [31].

#### 1.3.2. Advanced

Currently, laryngostroboscopy is employed experimentally both as a qualitative method to describe the phonatory movements of the vocal fold mucosa and for the purposes of quantitative measurements [32].

Since vocal onset time is reported to be as short as 36 ms, as shown by Orlikoff and colleagues in 2009 [33], laryngostroboscopy is too slow to show the vocal folds’ behavior until a steady oscillation state is attained. High-speed video laryngoscopy (HSV) has become the standard method for the detailed analysis of the voice onset, including the vocal fold attack time, as well as asymmetric or aperiodic vocal fold vibration. The first high-speed video attempts during indirect laryngoscopy were reported by Fransworth in 1940 [34]. Later, HSV was used during rigid endoscopy. Nowadays, transnasal endoscopy is preferred due to the lower levels of irritation and impairment experienced by the subjects under investigation. The image rates range from 1000 to 20,000 fps [35], allowing the comprehension of each vocal fold vibration’s amplitude, symmetry, and shape until a steady oscillation is reached. HSV was used by the research group around Johan Sundberg and Per-Åke Lindestad to investigate the triggering of vocal fold vibrations in aspirated and unaspirated productions of the consonant /p/ and in staccato tones [36].

From the video footage, the glottal area waveform (GAW) can be extracted, showing the opening and closing of the glottis over time, as described by Gomez and colleagues [37]. The GAW can be used for the calculation of the parameters of vocal fold dynamics [38]. Also, phonovibrograms (PVGs) can be extracted, visualizing the vocal folds’ movements for each glottal cycle, as described by Lohscheller and colleagues [39,40]. The resulting signals can be analyzed together with audio, electroglottographic, and flow or pressure signals [41,42]. The calculation of numeric parameters and their correlations provide multi-dimensional insights into laryngeal behavior in combination with airflow and acoustic outcomes.

Through this method, many findings have been added to voice onset research. Examples include longer initiation times, i.e., more preliminary cycles until a steady oscillation is reached, in older subjects [43]; pathological subjects [44,45]; or a higher sound pressure level, if required [46].

Future-orientated approaches already underway include improved automatic glottis segmentation to obtain the GAW in a shorter time, as well as the training of AI to detect voice disorders or typical patterns automatically [36,47,48].

### 1.4. Three-Dimensional Vocal Fold Oscillation

If one follows the body-cover theory of phonation as formulated by Minoru Hirano in 1974 [49], it becomes clear that vocal fold movements not only consist of an opening and closing movement in a horizontal line but also include three-dimensional deflections in the vertical.

Even if attempts are made to customize laser-based 3D visualizations for laryngoscopy [50], an important limitation to all endoscopic measures is the two-dimensional view of the vocal folds from above. The medial and inferior parts of the vocal folds remain invisible. The work carried out to compensate for this methodological limitation, starting from Ingo Titze’s work in 1988 [51], features the development of various computer-based models that take into account the three-dimensionality of vocal fold oscillations [52,53,54,55]. In these calculations, the medial surface and vertical thickness of the vocal folds were successfully allowed for [56].

### 1.5. Dynamic MRI for Imaging Voice Physiological Processes

Given that the larynx and vocal tract are anatomically connected, vocal tract adjustments may lead to changes in laryngeal configuration, as well as interactions that result from acoustic and aerodynamic coupling between the voice source and vocal tract. These interactions are taken into account in the current models [57].

Magnetic resonance imaging (MRI) is a suitable method for investigating the vocal tract not only in models but also in vivo. Improvements in the acquisition speed have made MRI technology a versatile tool for the study of voice physiology [58]. In voice research, dynamic MRI has been successfully applied to almost all organs involved in human voice production [59,60]. With static volumetric MRI of the vocal tract [61], acoustic properties could be simulated and 3D-printed vocal tracts generated for acoustic measurements [62,63].

The latest developments in our working group are designed to visualize the movements of the vocal folds using dynamic MRI [64]. The initial results are very encouraging, but it is not yet possible to use this technique to study vocal onset, as the technique—analogous to stroboscopic measurements—does not yet allow the dynamic onset processes of vocal fold oscillation to be imaged.

The number of data on laryngeal movements is expected to grow rapidly in the near future. This development could enable effective use to be made of artificial intelligence models, such as those already described for the identification of voice problems [65].

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
