# Peer review of "Investigation Methods for Vocal Onset—A Historical Perspective"

_bioengineering, 2024, doi:10.3390/bioengineering11100989_

Round 1

Reviewer 1 Report

Comments and Suggestions for Authors

Dear Authors,

Thank you very much for your manuscript, presenting an interesting issue, which is the historical development of investigation methods of vocal onset. Please pay attention to the following comments and queries, pertaining to your manuscript:

1.      Please let your manuscript be revised by a native English speaker. The linguistic flow of several sentences need to be optimized.

2.       From belief to knowledge: Please add some information regarding the increasing knowledge of larynx anatomy.

3.      General technical prerequisites: Please add any published contribution of artificial intelligence in understanding the pathophysiology of human voice and reproducing vocal onset.

Best Regards

Comments on the Quality of English Language

Moderate editing required

Author Response

  1. Please let your manuscript be revised by a native English speaker. The linguistic flow of several sentences need to be optimized.

We have had the manuscript revised again by a native English speaker to optimize the flow of language.

  1. From belief to knowledge: Please add some information regarding the increasing knowledge of larynx anatomy.

A paragraph on the history of laryngeal anatomy has been added (marked yellow).

  1. General technical prerequisites: Please add any published contribution of artificial intelligence in understanding the pathophysiology of human voice and reproducing vocal onset.

We have also added a sentence on the question of AI (marked yellow).These additions were backed up with 3 further references [9, 10 & 65] and the numbering of the entire literature was adjusted.

Reviewer 2 Report

Comments and Suggestions for Authors

- The manuscript is supposed to be a review article rather than a research article. 

- The merit of this manuscript is not clear. 

- Furthermore, there are no advanced scientific aspects provided or discussed in the manuscript in addition to the compilation of previous works. 

- Mathematical models and processes need to be clearly and completely provided and compared in detail. 

- Additional illustrations need to be provided in the manuscript to incorporate with the models and processes. 

- Conclusions including applications and utilization of the models and processes are missing from the manuscript. 

Author Response

- The manuscript is supposed to be a review article rather than a research article.

It is true that this manuscript is a review article and not an article with new data of its own.

 - The merit of this manuscript is not clear. 

- Furthermore, there are no advanced scientific aspects provided or discussed in the manuscript in addition to the compilation of previous works. 

- Mathematical models and processes need to be clearly and completely provided and compared in detail. 

This thematic focus, namely the historical presentation of voice research on the topic of vocal onset, was made at the special request of the Guest Editor Philippe Dejonckere.The value of the article lies precisely in the fact that, based on the historical derivation, the latest published research on the topic is compiled, including the work of our own research group (Highspeed glottography and dynamic MRI).

- Additional illustrations need to be provided in the manuscript to incorporate with the models and processes. 

- Conclusions including applications and utilization of the models and processes are missing from the manuscript. 

As this is a review article, it does not contain any illustrations. The models and procedures of the cited publications can be found in the respective original paper.

Round 2

Reviewer 1 Report

Comments and Suggestions for Authors

Dear Authors,

thank you for providing comprehensive and convincing answers to my questions and queries and made changes, that have contributed to the optimization of your manuscript and increased the publishing potential of your work. I have no further questions, pertaining to your manuscript.

Best Regards

Reviewer 2 Report

Comments and Suggestions for Authors

The revised manuscript is acceptable.